# Rethinking 3D-CNN in Hyperspectral Image Super-Resolution

Ziqian Liu, Wenbing Wang, Qing Ma, Xianming Liu and Junjun Jiang *

School of Computer Science and Technology, Harbin Institute of Technology, Harbin 150001, China; ziqian@stu.hit.edu.cn (Z.L.); wenbing.wang@rokid.com (W.W.); csmq@hit.edu.cn (Q.M.); csxm@hit.edu.cn (X.L.)
* Correspondence: jiangjunjun@hit.edu.cn

**Abstract:** Recently, CNN-based methods for hyperspectral image super-resolution (HSISR) have achieved outstanding performance. Due to the multi-band property of hyperspectral images, 3D convolutions are natural candidates for extracting spatial–spectral correlations. However, pure 3D CNN models are rare to see, since they are generally considered to be too complex, require large amounts of data to train, and run the risk of overfitting on relatively small-scale hyperspectral datasets. In this paper, we question this common notion and propose Full 3D U-Net (F3DUN), a full 3D CNN model combined with the U-Net architecture. By introducing skip connections, the model becomes deeper and utilizes multi-scale features. Extensive experiments show that F3DUN can achieve state-of-the-art performance on HSISR tasks, indicating the effectiveness of the full 3D CNN on HSISR tasks, thanks to the carefully designed architecture. To further explore the properties of the full 3D CNN model, we develop a 3D/2D mixed model, a popular kind of model prior, called Mixed U-Net (MUN) which shares a similar architecture with F3DUN. Through analysis on F3DUN and MUN, we find that 3D convolutions give the model a larger capacity; that is, the full 3D CNN model can obtain better results than the 3D/2D mixed model with the same number of parameters when it is sufficiently trained. Moreover, experimental results show that the full 3D CNN model could achieve competitive results with the 3D/2D mixed model on a small-scale dataset, suggesting that 3D CNN is less sensitive to data scaling than what people used to believe. Extensive experiments on two benchmark datasets, CAVE and Harvard, demonstrate that our proposed F3DUN exceeds state-of-the-art HSISR methods both quantitatively and qualitatively.

**Keywords:** 3D convolution; hyperspectral image; super-resolution; convolutional neural network





## 1. Introduction

Hyperspectral imaging systems receive reflections of multiple electromagnetic spectrum intervals from objects and acquire multi-band images. Thus, hyperspectral images (HSI) preserve the complete spectrum curve for each pixel, and this information is widely used in numerous areas, such as mineral exploration [1], medical diagnosis [2], and plant detection [3], etc. However, due to hardware limitations, most hyperspectral cameras must compromise between spatial resolution and spectral resolution. As a result, the spatial resolution of HSIs is relatively low compared to natural images, which becomes an obstacle to downstream high-level tasks such as image classification [4] and anomaly detection [5].

To solve this challenge, hyperspectral image super-resolution (HSISR) is in high demand, with the aim of reconstructing high-resolution (HR) MSIs given one or more inputs. Since the loss of spatial resolution comes from the multi-band property of an HSI, a natural idea is to fuse high-resolution multispectral images (HR-MSIs) and low-resolution hyperspectral images (LR-HSIs) to generate high-resolution hyperspectral outputs. Some works use corresponding RGB images as supplementary information [6–8], while others use panchromatic (PAN) images [9,10]. The main reason for the shortage of fusion-based methods is because co-registered HSI-MSI pairs are hard to collect, which limits their application scenarios. Therefore, we focus on an HSISR that is executed solely using LR-HSIs, i.e., single hyperspectral image super-resolution (SHISR), in this article.

Single hyperspectral image super-resolution is an ill-posed problem, as multiple high-resolution equivalents may correspond to a single low-resolution image. In the early period, this kind of method is highly dependent on handcrafted priors, such as low-rank [11] and sparsity [12]. However, these manually designed priors only reflect some aspects of hyperspectral data. Recently, convolutional neural networks (CNNs) have shown a strong representation ability, and they have been widely introduced to improve natural image super-resolution (SR) [13,14]. However, the mainstream of natural image SR methods is 2D CNN, which can lead to severe spectral distortion when applied to multi-band HSIs. Recently, some methods have emerged that combine hand-craft priors and neural networks and introduce mathematical modeling considering the properties of HSI, such as [15–17].

The problem caused by the high dimensionality in HSIs is severe. In [18], Cai et al. tried to solve this issue with a 3D attention mechanism and information bottleneck for classification tasks. However, in super-resolution, the model needs to preserve sufficient information of the input; thus, a natural way to solve this problem is to introduce 3D convolutions, which are good at extracting spatial–spectral correlations. The key difference between 3D and 2D convolutions is that 3D convolutions share parameters not only on spatial sides, but also on spectral dimension (see Figure 1). Thus, 3D convolutions can utilize the similarities between adjacent bands. The pioneering work of 3D CNN on HSISR is 3DFCNN [19], which consists of five cascaded 3D convolutional layers. However, the wave of full 3D CNN models did not last long, and 3D CNNs have received two main critiques. On the one hand, it is believed that 3D models are too complex to have good performance on HSI datasets with limited scale and are at risk of overfitting. On the other hand, some argue that the HSISR model should focus more on spatial enhancement, and 3D models cannot accomplish this. The latter comment leads to 3D/2D mixed models, which introduce extra 2D convolutions to boost spatial details. Representatives of such methods are MCNet [20] and ERCSR [21]. The former introduces multiple 2D branches to extract multi-scale spatial information from the features generated by 3D convolutions in each block (see Figure 2d), and the latter adds one 2D unit after every 3D unit to concentrate on spatial features (Figure 2c). Recently, graph-based methods have been applied to many tasks in hyperspectral image processing, such as classification [22–24], clustering [25–27], target detection [28], and anomaly detection [29], and have achieved good performance.

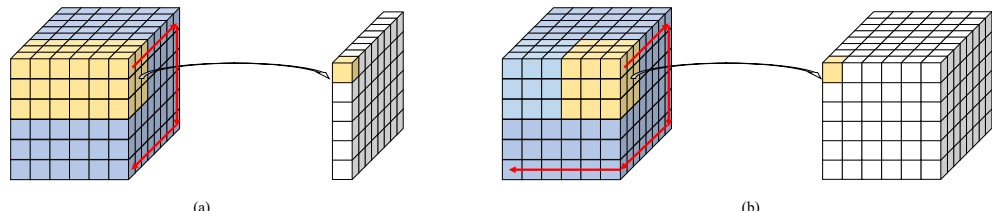

**Figure 1.** Comparison of 2D (**a**) and 3D convolution (**b**) on multi-band data.

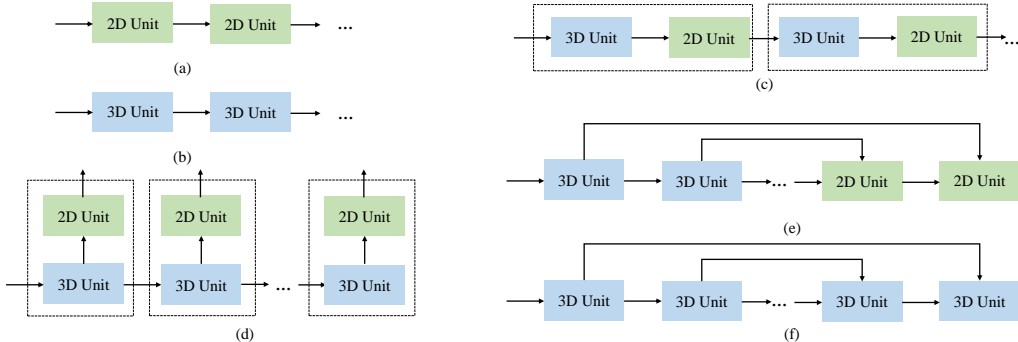

**Figure 2.** Simplified flowchart of several models: (**a**) full 2D CNNs, (**b**) full 3D CNNs, (**c**) ERCSR, (**d**) MCNet, (**e**) MUN, (**f**) F3DUN.

However, existing 3D HSISR models are very simple and plain, and they are not combined with some advanced inventions of deep learning. Therefore, it is worth questioning whether the critical views on 3D models still hold. We believe that the potential of 3D models has not been fully explored yet. This drives us to revisit those structures that have been proven effective in the super-resolution field and combine them with 3D CNN. Specifically, we propose a simple yet effective full 3D CNN model with a U-Net architecture, called Full 3D U-Net (F3DUN). F3DUN replaces the 2D blocks in normal U-Net with 3D convolutions and removes the downsampling–upsampling process in the middle. Experiments show that F3DUN outperforms state-of-the-art methods, which proves the effectiveness of full 3D models in HSISR tasks. Moreover, we design a 3D/2D mixed model that shares the same U-Net structure with F3DUN called Mixed U-Net (MUN), where cascaded 3D convolutions are used for feature extraction in the shallow half while 2D convolutions are gathered in the deep half for spatial enhancement. F3DUN and MUN provide an excellent pair to compare the two kinds of model priors, 3D/2D mixed models and full 3D CNN models. In Figure 2, we summarize the main model priors in HSISR. The experimental results show that full 3D CNN models have larger modeling capacities than the 3D/2D mixed model with the same number of parameters; that is, full 3D CNN models have better performance with sufficient training data. Another common concern regarding full 3D CNN models is that they cannot perform well on small-scale datasets, since they demand a lot of training samples. A comparison on a small-scale dataset disproves this point, since F3DUN still obtains competitive results with MUN. With little harm on performance, F3DUN eliminates severe overfitting. Thus, we conclude that full 3D CNN models are less sensitive to the scale of training sets than we used to think and that overfitting can be prevented with carefully designed architectures.

In summary, the contributions we made in this paper are listed here:

- We rethink the role that 3D CNN plays in the HSISR field and design a novel full 3D CNN model based on thd U-Net architecture called Full 3D U-Net (F3DUN). Experimentally, it outperforms existing state-of-the-art, single-image SR methods, which proves the effectiveness of full 3D CNN models in this field.
- We develop a mixed 3D/2D model that shares the same structure with F3DUN, termed Mixed U-Net (MUN), for comparison. Extensive analysis on the two models shows that the full 3D CNN model has a larger modeling capacity than the 3D/2D mixed model with the same number of parameters; thus, it performs better with large-scale datasets.
- We explore the relationship between the scale of training samples and the prior of the model. We argue that the full 3D CNN model can obtain competitive results on small-scale training sets compared with the 3D/2D mixed model, which concludes that the full 3D CNN model is more robust with respect to the amount of training samples than commonly thought.

The rest of this paper is organized as follows: In Section 2, we briefly introduce related works on hyperspectral and natural image super-resolution. Then, in Section 3, we describe the model designs of F3DUN and MUN in detail. Experiments on public datasets are shown and discussed in Section 4, and the conclusion is in Section 5. For ease of reading, we have provided a list of abbreviations in Table 1 for reference.

**Table 1.** Abbreviations used in the paper.

| Abbreviation | Full Name |
|---|---|
| HSI | hyperspectral image |
| LR | low-resolution |
| HR | high-resolution |
| SR | super-resolution |
| HSISR | hyperspectral image super-resolution |
| SHISR | single-image hyperspectral super-resolution |
| MUN | Mixed U-Net |
| F3DUN | Full 3D U-Net |

## 2. Related Work

In this section, we briefly review some research fields related to our work. First, we introduce 3D convolution and its applications in computer vision tasks. Then, we give a big picture of hyperspectral image super-resolution, including single hyperspectral image super-resolution and fusion-based hyperspectral image super-resolution. At the end of this section, the development of single natural image super-resolution is summarized.

### 2.1. Three-Dimensional Convolution

Three-dimensional convolution shares parameters not only on spatial dimensions but also on channel dimension, which is suitable to deal with cube data such as HSIs or video. A typical application of 3D convolution is hyperspectral image classification [30–33], since 3D convolution can tackle the rich spatial–spectral information contained in HSIs. Furthermore, Jiang et al. [34] used a 3D convolutional neural network for dehazing in HSIs. The application of 3D convolution expands to some other remote sensing data, such as SLAM [35] and LiDAR [36,37]. Three-dimensional convolution also plays an important role in video processing due to its ability to capture inter-frame relationships. Li et al. [38] proposed a 3D CNN for video person re-identification, and Ying et al. [39] employed deformable 3D convolution in a video super-resolution task.

### 2.2. Single Hyperspectral Image Super-Resolution

Single hyperspectral image super-resolution (SHISR) aims to reconstruct an HR-MSI solely from its LR opponent. Without the need of an auxiliary image, SHISR can be widely applied and thus attracts considerable research attention. The pioneering work is [40], in which Akgun et al. proposed an acquisition model and applied the projection onto convex sets (POCS) algorithm [41] on SHISR. Subsequently, Huang et al. [12] introduced low-rank and sparsity constraints on dictionary learning to reconstruct HR-MSIs. The aforementioned works can be categorized as model-driven methods, since they rely on manually designed priors. With the coming of the deep learning era, data-driven frameworks are also being applied in the SHISR field. Yuan et al. [42] tried to transfer knowledge from natural image SR to SHISR and combined CNN and non-negative matrix factorization (NMF) to super-resolve MSIs. Xie et al. [43] utilized CNN and NMF to extract spatial and spectral features separately, super-resolving HSIs in a non-end-to-end manner. Due to the strong representative ability, Mei et al. [19] first proposed a five-layer full 3D CNN (3D-FCNN) for SHISR. Subsequently, Li et al. [20] designed a mixed 2D/3D CNN (MCNet), introducing multiple 2D branches to extract multi-scale spatial information. Furthermore, in [21], the authors replaced the redundant multi-branch structure with interleaved 2D/3D blocks. On the other hand, Li et al. [44] proposed a full 2D CNN model (GDRRN) with grouped convolutions and recursive architecture. Then, Jiang et al. [45] divided HSIs into overlapped band groups based on the similarity of adjacent bands, and designed a multi-branch model (SSPSR) in a divide-and-merge manner. Li et al. [46] presented intra-group and inter-group fusion to gather features from different band groups. Another category of SHISR methods is sequence models, motivated by video SR. Wang et al. [47] proposed a dual-channel model, fusing spatial and spectral features from neighboring

bands and super-resolving LR-MSIs band by band. Recently, Liu et al. [48] proposed a parallel network called Interactformer for SHISR, which contains a Transformer branch and a 3D-CNN branch; the Transformer branch is used to capture long-range dependencies to obtain global features, and the 3-D CNN branch is used to extract local features while preserving the spectral correlation of HSIs.

### 2.3. Fusion-Based Hyperspectral Image Super-Resolution

Fusion-based hyperspectral image super-resolution is a classical method of HSISR with a long history. Its basic idea is to fuse the information of LR-HSIs and HR-MSIs to improve the spatial resolution of the LR-HSIs. Yokoya et al. [49] proposed an approach based on coupled non-negative matrix factorization (CNMF). In addition, more researchers have taken the abundant spectral information encoded in HSIs into consideration and have exploited various priors, such as sparsity [50,51], non-local similarity [52], tensor and low-rank constraints [53]. Recently, deep learning methods have shown their superiority in feature representation and they rely less on assumptions; thus, they have gradually become popular [54]. Some deep unfolding networks have been proposed, which turn a mathematical model to a neural network [6,7,55,56]. However, all the above methods require co-registered HSI-MSI pairs, which severely limits the scope of their application, since the collection of paired data can be extremely hard.

### 2.4. Single Natural Image Super-Resolution

Single-image super-resolution (SISR) has been a research topic for a long time. In the early stages, researchers focused on designing good priors to super-resolve RGB or gray images [57–59]. With the rise of deep learning, CNN-based SISR methods have played an important role. Dong et al. [60] proposed the first end-to-end model consisting of three convolutional layers. Later, Kim et al. [13] presented a very deep convolutional model of 20 layers for SISR, and Lim et al. [14] introduced residual connection and removed unnecessary modules to make the model more effective and robust. Afterwards, enormous CNN-based works emerged, such as RCAN [61], SAN [62], HAN [63], NLSA [64], IPT [65], and SwinIR [66], which achieved considerable success.

## 3. Method

In this section, we describe the details of the proposed Full 3D U-Net (F3DUN). Our main interest in F3DUN is whether the performance of previous full 3D CNN models is limited by their inherent nature or by flaws in the designs of their architectures. We also introduce F3DUN's 3D/2D mixed counterpart, Mixed U-Net (MUN), in order to analyze the effect of 3D convolutions in HSISR models.

### 3.1. Full 3D U-Net

F3DUN follows the idea of a full 3D CNN that can be traced back to 3DFCNN and, at the same time, integrates advanced experience in model design. As is shown in Figure 3, our full 3D U-Net consists of three parts: shallow feature extraction, nonlinear mapping, and an upsampler. To prevent memory loss and generate multi-scale features, multiple skip connections are introduced.

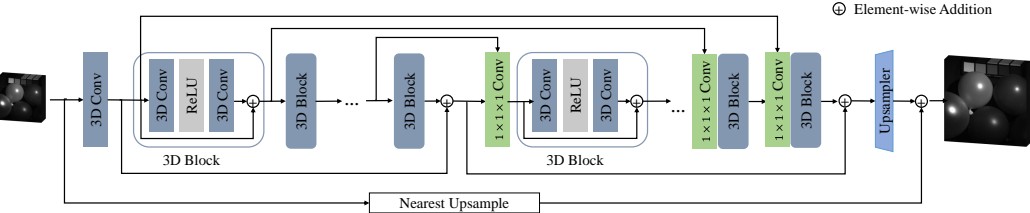

**Figure 3.** Overview of F3DUN.

For clarity, let $I_{LR} \in \mathbb{R}^{L \times H \times W}$ and $I_{HR} \in \mathbb{R}^{L \times sH \times sW}$ denote the LR-MSI and HR-MSI, respectively, where $H$ and $W$ represent the height and weight of each band, $L$ is the number of spectrum bands, and $s$ indicates the scale factor. The super-resolved HSI has the same size of $I_{HR}$ and is remarked as $I_{SR}$. Given $I_{LR}$ as input, we first use a 3D convolution of $3 \times 3 \times 3$ to preliminarily extract information:

$$F_{3D}^0 = Conv_{3D}^0(unsqueeze(I_{LR})). \tag{1}$$

The aim of the unsqueeze operation is to transfer $I_{LR}$ to size of $1 \times L \times H \times W$ so that it can be handled by 3D convolution.

Then, the shallow feature $F_{3D}^0$ is sent to the nonlinear mapping part, which consists of cascaded 3D blocks. Each 3D block in the shallow half of this part has two $3 \times 3$ convolutions and a ReLU activation function in between. In addition, all 3D blocks use a residual connection for stability, and its formula can be written as:

$$F_{3D}^k = Block_{3D}^k(F_{3D}^{k-1}) + F_{3D}^{k-1} \tag{2}$$
$$= Conv_{3 \times 3 \times 3}(ReLU(Conv_{3 \times 3 \times 3}(F_{3D}^{k-1}))) + F_{3D}^{k-1}. \tag{3}$$

We use skip connections to link 3D blocks at symmetrical locations in order to prevent memory loss and provide cross-scale information for deep 3D blocks:

$$F_{3D}^{K-k} = Block_{3D}^{K-k}([F_{3D}^{K-k}, F_{3D}^k]) \quad for \quad k = 1, 2, ..., K/2, \tag{4}$$

where $K$ is the total number of 3D blocks for non-linear mapping, and $[\cdot, \cdot]$ denotes concatenation operation. In addition, 3D blocks in the deep half add an extra $1 \times 1 \times 1$ convolution to fuse the information from the previous block and the skip connection. Beyond skip connections within the U-Net architecture, we further introduce three long-distance connections. Two of them are located in the first and second halves of the non-linear mapping part:

$$F_{3D}^{'K/2} = F_{3D}^{K/2} + F_{3D}^0, \tag{5}$$
$$F_{3D}^{'K} = F_{3D}^{'K/2} + F_{3D}^K, \tag{6}$$

and the other one is a global skip connection:

$$I_{SR} = Upsampler(F_{3D}^{'K}) + nearest_u psample(I_{LR}). \tag{7}$$

As for the upsampler, the input is $F_{3D}^{'K}$, and we use a transposed 3D convolutions to scale it up to $b \times C \times L \times sH \times sW$. At last, a pseudo-3D convolution, which is made of a $1 \times 3 \times 3$ and a $3 \times 1 \times 1$ filter, squeezes the channel number to 1. For $4\times$ super-resolution, we directly use one transposed 3D convolution layer for upsampling. When the scale factor is 8, we first use one transposed 3D convolution layer to upsample to $4\times$, and then use another transposed 3D convolution layer to scale up to the desired size. This progressive upsampling strategy prevents the upsampling factor from being too large beyond the capabilities of a single deconvolution layer.

### 3.2. Mixed U-Net

The basic idea of Mixed U-Net (MUN) is to divide the non-linear mapping of the network into two parts: a group of 3D convolutions for feature extraction and cascaded 2D convolutions for spatial enhancement. The specialized structure of MUN can effectively avoid the feature confusion problem caused by interleaved structures of 3D and 2D modules in previous 3D/2D mixed models, and skip connections can also effectively suppress memory forgetting. MUN provides a useful tool to compare 3D/2D mixed models and full 3D CNN models, since it shares similar architecture with F3DUN.

As is shown in Figure 4, MUN shares the same architecture with F3DUN, only replacing 3D Blocks with 2D blocks in the deep half of the non-linear part. The 2D blocks follow

the structure of a resblock and use a $1 \times 1$ convolution to fuse the two inputs from the previous block and the skip connection, respectively. The pipeline of the 2D block in MUN can be depicted as:

$$F_{2D}^k = Block_{2D}([F_{2D}^{k-1}, reshape(F_{3D}^{K-k}]), \tag{8}$$

where $K$ is the total number of 3D and 2D blocks. Here, 3D features $F_{3D}^{K-k}$ are reshaped from $b \times C \times L \times H \times W$ to $(b \times L) \times C \times H \times W$ before being fed into the corresponding 2D block, where b and C denote batch size and feature channels, respectively. At the junction of the 3D group and the 2D group, i.e., after the $N$th 3D block, a particular 2D block is added without the fusion convolution.

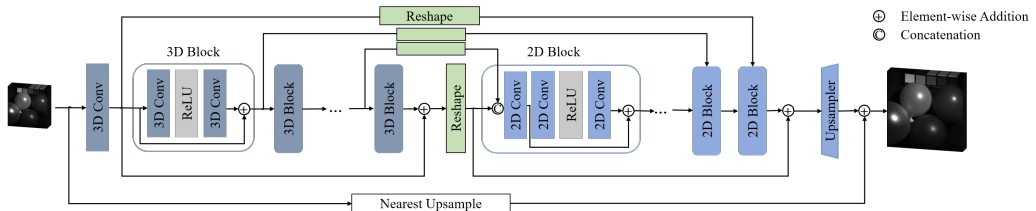

**Figure 4.** Overview of MUN.

*3.3. Loss Function*

Following the main stream of HSISR methods, we employ L1 loss to lead the model to generate sharp results:

$$L_{rec} = \frac{1}{M} \sum_{i=1}^{M} ||I_{SR}^i - I_{HR}^i||_1, \tag{9}$$

where $M$ denotes the number of samples. However, in practice, we find that although reconstruction loss can effectively constrain spatial similarity, spectral distortion is still a serious problem. Here, we additionally introduce a loss based on the SAM [67] criterion for HSI quality assessment to constrain the spectral similarity:

$$L_{SAM} = \frac{1}{sH \times sW} \sum_{(x,y)} \arccos\left(\frac{< I_{SR}^{(x,y)}, I_{HR}^{(x,y)} >}{||I_{SR}^{(x,y)}||_2 ||I_{HR}^{(x,y)}||_2}\right), \tag{10}$$

where $< \cdot, \cdot >$ means dot production and $(x, y)$ denotes the spatial coordinate of the pixel. Since the derivative of $\arccos(x)$ is $-\frac{1}{\sqrt{1-x^2}}$, there exists a gradient explosion problem when $x^2 \to 1$. Therefore, we filter the pair of pixels whose cosine similarity lies in $[-1 + \epsilon, 1 - \epsilon]$ and only apply SAM loss on them. In practice, we choose $\epsilon = 10^{-8}$. In general, the total loss function is:

$$L_{total} = Lrec + \lambda L_{SAM}, \tag{11}$$

in which $\lambda$ is a hyperparameter to balance the two losses. In practice, we set $\lambda = 0.05$. This loss function is applied to both F3DUN and MUN.

*3.4. Relationship with Other Methods*

(a) 3DFCNN: 3DFCNN is the pioneering work of full 3D CNN models for HSISR. It consists of 5 3D convolution layers and uses MSE loss to train. However, there are noticeable drawbacks to it. Most importantly, the model architecture of 3DFCNN is too shallow and it does not combine with advanced ideas of deep learning, such as residual learning and long skip connections. Therefore, in this paper, we rethink the prior of full 3D CNN in HSISR and argue that full 3D CNN models' bad results are not caused by 3D CNN, but rather a lack of model design. It is obvious that F3DUN is much deeper than 3DFCNN and successfully prevents overfitting and achieves SOTA results.

(b) Two- and three-dimensional mixed models: The main idea of 3D/2D mixed models is to introduce 2D convolutions into 3D CNN in order to boost the ability of spatial

enhancement. They have become popular recently and can be seen as a modification to full 3D CNN models. However, there are two potential problems. On the one hand, the 2D convolutions in 3D/2D models are shared in the spectral dimension, which leads to a risk of spectral distortion. On the other hand, 3D/2D mixed models always face the dilemma of balancing the two kinds of convolutions, where features enhanced by 2D convolutions can be polluted by cascading 3D modules. In MUN, we partially solve the two above problems, but we make a step forward: could a full 3D CNN model outperform 3D/2D mixed ones? Model analysis on MUN and F3DUN supports our idea and proves the advantages of the full 3D CNN model compared with 3D/2D mixed models.

## 4. Experiments and Analysis

In this section, we conduct comprehensive experiments to evaluate the effectiveness of the proposed F3DUN. Two benchmark datasets, i.e., CAVE [68] and Harvard [69], are used for the comparisons. We present the quantitative and perceptual results of our F3DUN together with seven existing HSISR methods, including Bicubic, EDSR [14], 3DFCNN [19], SSPSR [45], MCNet [20], ERCSR [21], and SFCSR [47]. Furthermore, analysis between F3DUN and MUN using different data scales reveals the data efficiency of our proposed method.

### 4.1. Datasets

(a)  CAVE dataset: The CAVE dataset [68] is taken by a cooled CCD camera, and the range of the wavelength is from 400 nm to 700 nm, at a step of 10 nm (31 bands). The 32 hyperspectral images are divided into five sections: real and fake, skin and hair, paints, food and drinks, and stuff. Each image has a size of $512 \times 512 \times 31$, and every band is stored as a grayscale picture separately in the form of a PNG.

(b)  Harvard dataset: The Harvard dataset [69] contains 77 indoor and outdoor hyperspectral images under daylight illumination collected by the Nuance FX, CRI Inc., camera. Each image covers the wavelength range of 400 nm to 700 nm, evenly divided into 31 bands. The spatial resolution is $1040 \times 1392$, and all images are stored as .mat files.

### 4.2. Implementation Details

For the CAVE dataset, we select 12 images as the test set, and the remaining 20 images are used for training. When the scale factor is 4, each training image is cropped into $96 \times 96$ patches with a stride of 48 pixels. For an $8\times$ super-resolution, the patch size and stride are $192 \times 192$ and 36. After that, we use bicubic downsampling to generate LR patches and LR images for test. 10% of the cropped patches are randomly sampled to form a validation set. Several data argumentation methods are applied, including rotations of $90°$, $180°$ and $270°$, vertical and horizontal flips, and their combinations.

The training samples of the Harvard dataset are cropped randomly on the spatial dimensions, and each training image generates 72 patches for both $4\times$ and $8\times$ super-resolution. The patch sizes of the training samples are $96 \times 96$ and $128 \times 128$ for scale factors of 4 and 8, respectively. Additionally, we select 5 and 3 images from the outdoor and indoor categories as test set. Due to the image size in the Harvard dataset being too large, we crop the top-left $1024 \times 1024$ part of each test image and divide it into 4 sub-images of $512 \times 512$. Then the LR images are generated by bicubic downsampling.

We trained F3DUN for scale factors of 4 and 8, with random initialization. The model contains 10 and 12 3D blocks for $4\times$ and $8\times$ super-resolutions, respectively, and the number of filters in each convolutional layer is set to 64. We adopt the Adam optimizer [70] and set $\beta_1 = 0.9$ and $\beta_2 = 0.999$. The learning rate is initialized as $10^{-4}$ and halved every 35 epochs. We employ the batch size to be 16 and train our model for 200 epochs. Our algorithm is performed using PyTorch library with NVIDIA GeForce GTX 2080ti GPUs.

### 4.3. Evaluation Metrics

We adopt six common metrics to evaluate the quality of super-resolved images: peak signal-to-noise ratio (PSNR), structure similarity (SSIM) [71], cross correlation (CC) [72], spectral angle mapper (SAM) [67], root mean squared error (RMSE), and erreur relative globale adimensionnelle de synthese (ERGAS) [73]. We compute the mean of all bands for PSNR and SSIM, which are generally used to measure image restoration quality quantitatively:

$$PSNR = \frac{1}{L} \sum_{l=1}^{L} 10 \log_{10}\left(\frac{MAX_l^2}{MSE_l}\right), \tag{12}$$

$$SSIM = \frac{1}{L} \sum_{l=1}^{L} \frac{(2\mu_{I_{SR}}^l \mu_{I_{HR}}^l + c_1)(2\sigma_{I_{SR}}^l \sigma_{I_{HR}}^l + c_2)}{mn}, \tag{13}$$

where

$$m = (\mu_{I_{SR}}^l)^2 + (\mu_{I_{HR}}^l)^2 + c_1, \tag{14}$$

$$n = (\sigma_{I_{SR}}^l)^2 + (\sigma_{I_{HR}}^l)^2 + c_2. \tag{15}$$

Here, $L$ denotes the total number of bands of the hyperspectral image, and will refer to the same hereinafter. SAM is widely used to compare the spectral difference between two hyperspectral images, and its definition can be found in Equation (10). RMSE measures the absolute error between the reconstructed image and the ground truth. CC and ERGAS are often employed in HS fusion, and they reflect the overall quality of the reconstructed images. Their mathematical definitions are as follows:

$$CC = \frac{1}{L} \sum_{l=1}^{L} \frac{cov\left(I_{SR}^l, I_{HR}^l\right)}{\sigma_{I_{SR}^l} \sigma_{I_{HR}^l}}, \tag{16}$$

$$ERGAS = 100s \sqrt{\frac{1}{L} \sum_{l=1}^{L} \left(\frac{RMSE_l}{mean\left(I_{HR}^l\right)}\right)^2}. \tag{17}$$

Here, $s$ represents the scale factor, and *mean* denotes the average operation. The limit values of the six above indicators are $+\infty$, 1, 0, 0, 0, and 1, respectively.

### 4.4. Model Analysis

4.4.1. Comparison with 3D-2D Model Prior

In this section, we perform a comparison on F3DUN and MUN to analyze the advantages and disadvantages of the 3D/2D mixed model and the full 3D CNN model for the HSISR task. Without loss of generality, this model analysis is performed on the CAVE dataset, and the scale factor is set to 4. And we made comparisons on both normal and reduced datasets, in order to explore the relationship between data scale and model priors. The results can be found in Tables 2 and 3 respectively. The reason why MUN is chosen as a representative of 3D/2D mixed model is manifold. Firstly, MUN has been proven to outperform other HSISR models, especially all previous 3D/2D mixed models (see Tables 2 and 4). Secondly, F3DUN and MUN share similar architectures, making the comparison fairer and more significant. Thirdly, it is easy to control the number of parameters of the two models so that they are on the same level.

**Table 2.** Comparison of the full 3D CNN model and the 3D/2D mixed model on the **original** CAVE dataset.

| Method | Param. | PSNR↑ | SSIM↑ | RMSE↓ | SAM↓ | ERGAS↓ | CC↑ |
|---|---|---|---|---|---|---|---|
| MUN | 2.4 M | 36.8860 | 0.9452 | 0.0188 | 3.6500 | 5.4975 | 0.9779 |
| F3DUN | 2.5 M | **36.9368** | **0.9458** | **0.0186** | **3.6340** | **5.4429** | **0.9786** |

**Table 3.** Comparison of the full 3D CNN model and the 3D/2D mixed model on the **reduced** CAVE dataset.

| Method | Param. | PSNR↑ | SSIM↑ | RMSE↓ | SAM↓ | ERGAS↓ | CC↑ |
|---|---|---|---|---|---|---|---|
| MUN | 2.4M | 36.7015 | **0.9454** | 0.0189 | **3.6472** | 5.5522 | **0.9783** |
| F3DUN | 2.5M | **36.7249** | 0.9453 | **0.0188** | 3.6496 | **5.5330** | **0.9783** |

In Table 2, we summarize the quantitative results of the two models on the original CAVE dataset. We control the two models to have roughly the same amount of parameters, thereby ruling out performance gains for one model due to having more parameters. It can be seen that F3DUN beats MUN on all indicators, excluding SSIM, which indicates that the full 3D CNN model has a larger modeling capacity than the 3D/2D model. Considering that the main difference between F3DUN and MUN is the use of 3D convolution and 2D convolution in the deep half of the nonlinear mapping, we can reason that the advantage of F3DUN is basically brought about by 3D convolutions. In Figure 5, we present a visual comparison between MUN and F3DUN, and F3DUN shows some superiority in edge and complex context recovery tasks.

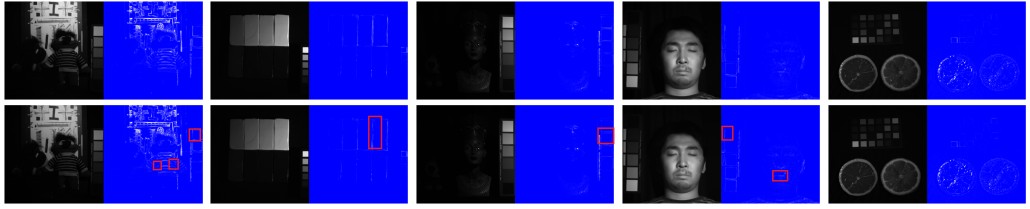

**Figure 5.** Some visual results and error maps of MUN (the upper line) and F3DUN (the bottom line).

A long-standing concern about full 3D CNN models is that they are computationally more complex and therefore prone to poor results due to overfitting on hyperspectral image datasets, the scale of which is limited by various sensors. This concern follows common sense in the field of deep learning. However, we want to more precisely analyze the conditions in which the full 3D CNN model could work. To further investigate this issue, we adjust the step size of patch cropping in the CAVE dataset to 72 pixels and thus construct a smaller training set. In the original training set, each image generates 81 patches, while only 36 patches are collected per image in the reduced training set.

The performances of MUN and F3DUN on the reduce dataset are shown in Table 3. With fewer training samples, both models have a certain decline in most indicators, which is consistent with the common sense of deep learning. However, the full 3D CNN model still achieves acceptable results on this smaller dataset, even outperforming the 3D/2D mixed model on several metrics such as PSNR, RMSE, and ERGAS. This demonstrates that the full 3D CNN model is less sensitive to the scale of training samples, contrary what people commonly thought, and it could achieve competitive performance compared with the 3D/2D mixed model on the small-scale dataset. In addition, if we put the two tables together, it suggests that the full 3D CNN model is more effective than the 3D/2D mixed model when sufficiently trained.

### 4.4.2. Results on Noisy Images

Hyperspectral images are prone to suffering from various distortions. However, the robustness to degradation is not our main concern. On the one hand, we argue that benchmarks are collected by real-world sensors, so a model's performance on them is meaningful for practical cases. On the other hand, super-resolution is a subset of hyperspectral image reconstructions, and there are many techniques that deal with degradation, such as denoising [74,75] and deblurring [76,77].

In Figures 6 and 7, we present reconstruction results with noisy inputs on the CAVE dataset. It can be seen that our model is somewhat robust to Gaussian noise, but can be severely affected by impulse noise. The main reason for this is that we do not introduce noise manually in training, and the model has no chance to learn to remove it.

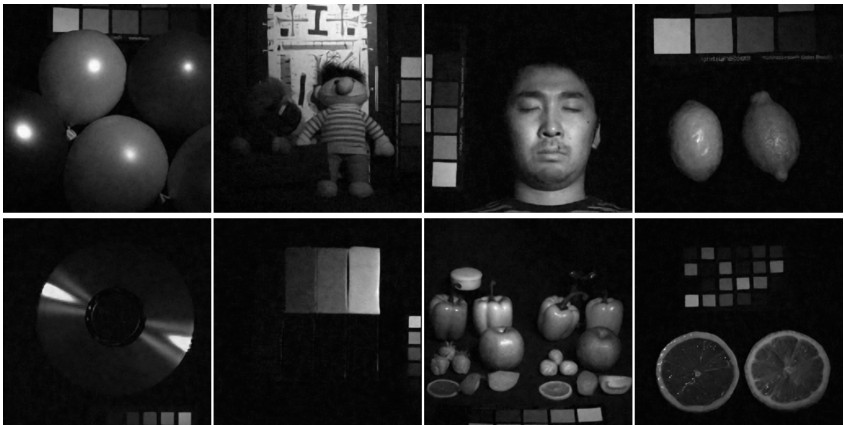

**Figure 6.** Some reconstruction results with Gaussian noise ($\sigma = 5$) on the CAVE dataset ($4\times$).

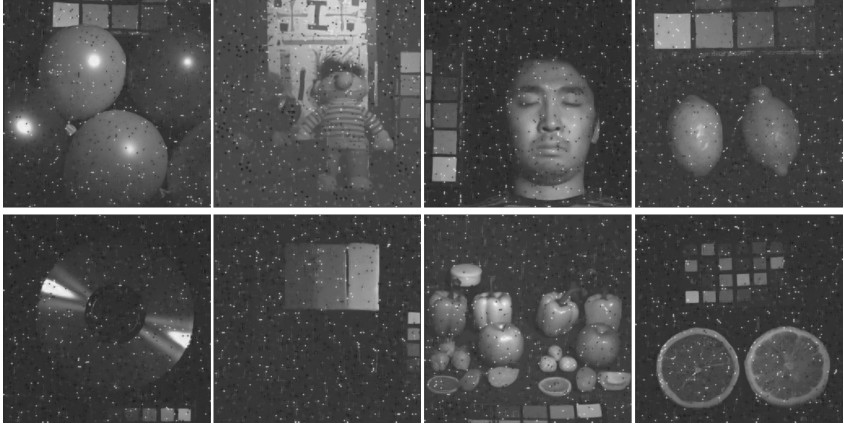

**Figure 7.** Some reconstruction results with impulse noise (3%) on the CAVE dataset ($4\times$).

### 4.5. Comparison with State-of-the-Art Methods

In this section, we compare the proposed F3DUN with several existing single-image hyperspectral super-resolution methods. The selected methods widely cover popular model priors in this field. Among them, bicubic upsampling is a traditional interpolation-based super-resolution approach, and EDSR [14] is a full 2D convolutional neural network that achieves remarkable success in natural image super-resolution. The 3DFCNN [19] model is the pioneer work that applies the full 3D CNN model to the HSISR task. Moreover, SSPSR [45] is a full 2D model designed for HSISR specifically using the similarity of neighboring bands. Another kind of method is 3D/2D mixed models, including MCNet [20] and ERCSR [21]. Finally, SFCSR [47] is a sequence model that treats bands in an HSI as frames in a video.

A quantitative comparison on the CAVE dataset is shown in Table 4. It can be seen that our method outperforms other models on all six metrics for both $4\times$ and $8\times$ super-

resolutions. Bicubic upsampling obtains the worst results in both cases, since it cannot recover missing high-frequency components in LR-MSIs. The comparison between 3DFCNN and F3DUN supports our hypothesis: the full 3D CNN model can achieve SOTA performance by carefully designing the network structure. A 3D/2D mixed model, MCN, achieves the second highest score on PSNR for a 4× scale factor, but it fails to generate good SR images when the scale factor increases to 8. In contrast, ERCSR performs better on 8× upsampling. The sequence model SFCSR obtained results second only to F3DUN in most of the indicators of the 8× super-resolution experiment, showing good reconstruction ability and balance in several indicators. However, the advantage of our model is obvious, proving the effectiveness of full 3D CNN in HSISR. In the 4× super-resolution experiment, F3DUN outperforms the advanced 3D/2D mixed model MCNet by 0.22dB on the PSNR metric. Furthermore, in the case of 8×, F3DUN is 0.27dB higher than the second best model on the PSNR index and significantly outperforms other methods in the SSIM index, which illustrates that F3DUN is more robust and capable for larger scale factors. Additionally, there are clear gains on the SAM score, namely, −0.28 rad for 4× and −0.45 rad for 8×, showing the remarkable effect of introducing SAM loss on suppressing spectral distortion.

**Table 4.** Quantitative evaluation of different HSISR methods on the CAVE dataset. The best and second best results are in **bold** and underlined, respectively.

| Method | Scale | PSNR↑ | SSIM↑ | RMSE↓ | SAM↓ | ERGAS↓ | CC↑ |
|---|---|---|---|---|---|---|---|
| Bicubic | 4 | 34.6163 | 0.9287 | 0.0225 | 5.1869 | 6.8047 | 0.9728 |
| EDSR [14] | 4 | 36.1569 | 0.9393 | 0.0202 | 4.4149 | 5.9793 | 0.9758 |
| 3DFCNN [19] | 4 | 35.3479 | 0.9349 | 0.0209 | 4.0415 | 6.0750 | 0.9758 |
| SSPSR [45] | 4 | 36.5934 | 0.9431 | 0.0195 | 3.9691 | 5.7608 | 0.9766 |
| MCNet [20] | 4 | 36.7182 | 0.9434 | 0.0191 | 3.9590 | 5.6281 | 0.9769 |
| ERCSR [21] | 4 | 36.6318 | 0.9436 | 0.0193 | 3.9140 | 5.6610 | 0.9772 |
| SFCSR [47] | 4 | 36.6718 | 0.9438 | 0.0191 | 3.9218 | 5.6409 | 0.9774 |
| F3DUN | 4 | **36.9368** | **0.9458** | **0.0186** | **3.6340** | **5.4429** | **0.9786** |
| Bicubic | 8 | 30.0719 | 0.8511 | 0.0367 | 7.0948 | 10.8664 | 0.9349 |
| EDSR [14] | 8 | 30.9524 | 0.8703 | 0.0347 | 6.0270 | 10.1562 | 0.9393 |
| 3DFCNN [19] | 8 | 30.3622 | 0.8669 | 0.0354 | 5.6714 | 10.2498 | 0.9391 |
| SSPSR [45] | 8 | 30.4524 | 0.8670 | 0.0367 | 5.6093 | 10.7448 | 0.9322 |
| MCNet [20] | 8 | 31.2193 | 0.8790 | 0.0339 | 5.6056 | 9.7672 | 0.9403 |
| ERCSR [21] | 8 | 31.3491 | 0.8808 | 0.0335 | 5.5721 | 9.7204 | 0.9421 |
| SFCSR [47] | 8 | 31.4008 | 0.8814 | 0.0330 | 5.6069 | 9.6230 | 0.9432 |
| F3DUN | 8 | **31.6730** | **0.8849** | **0.0325** | **5.1230** | **9.4327** | **0.9441** |

The results of the Harvard dataset are summarized in Table 5. The salient feature of the Harvard dataset is that the training samples are sparser than the CAVE dataset, since they are randomly cropped and possibly unable to cover the entire training images. Under these conditions, our F3DUN still generates competitive SR results. It can be seen that SSPSR achieves sub-optimal scores for most indicators on the Harvard dataset with respect to 4× SR. SSPSR shows slight advantage on SAM, and this margin possibly comes from its divide-and-merge strategy. Since SSPSR first processes each band separately and then merge all bands together, this architecture can preserve spectral information. Moreover, F3DUN is optimal on all six indices. When the scale factor is 8, our method achieves the highest scores, except in RMSE and SAM. Specifically, F3DUN performs best on the SSIM and ERGAS metrics, which indicates that the reconstructed images preserve more complete structures and have higher overall quality. The lightweight 3D/2D mixed model ERCSR obtains the second-best scores on most indices, and the sequence model SFCSR achieves pretty good results, which suggests the advantages of models with less complexity. However, with the help of a deliberately designed architecture, our model still obtains an outstanding output.

In Table 6, we compare the model scale and computational complexity of our F3DUN and other state-of-the-art methods. It is shown that our model achieves superior performance with little extra computational cost. In specific, the number of parameters and FLOPs of F3DUN are at the same level of MCNet, while F3DUN outperforms MCNet on all six quantitative metrics. In addition, the main cost of our model comes from its pure 3D CNN structure, since 3D convolutions are very dense in terms of parameter and calculations. Our model is slightly heavier than the mixed 3D/2D models (i.e., MCNet and ERCSR), and 3DFCNN is lightweight due to its shallow structure. Though sequence models such as SFCSR have fewer parameters, they can run slowly due to their bad parallelism.

**Table 5.** Quantitative evaluation of different HSISR methods on the Harvard dataset. The best and second best results are in **bold** and underlined, respectively.

| Method | Scale | PSNR↑ | SSIM↑ | RMSE↓ | SAM↓ | ERGAS↓ | CC↑ |
|---|---|---|---|---|---|---|---|
| Bicubic | 4 | 42.4128 | 0.9265 | 0.0137 | 3.0520 | 3.4897 | 0.9519 |
| EDSR [14] | 4 | 43.0512 | 0.9359 | 0.0126 | 2.7879 | 3.2591 | 0.9578 |
| 3DFCNN [19] | 4 | 42.8021 | 0.9310 | 0.0129 | 2.8593 | 3.3495 | 0.9554 |
| SSPSR [45] | 4 | <u>43.4522</u> | <u>0.9387</u> | **0.0120** | <u>2.6788</u> | <u>3.0980</u> | <u>0.9615</u> |
| MCNet [20] | 4 | 43.3126 | 0.9375 | 0.0124 | 2.7428 | 3.1590 | 0.9602 |
| ERCSR [21] | 4 | 43.2983 | 0.9376 | 0.0124 | 2.7416 | 3.1839 | 0.9600 |
| SFCSR [47] | 4 | 43.3817 | 0.9383 | <u>0.0123</u> | 2.7571 | 3.1462 | 0.9605 |
| F3DUN | 4 | **43.4855** | **0.9389** | **0.0120** | **2.6555** | **3.0923** | **0.9618** |
| Bicubic | 8 | 38.3909 | 0.8667 | 0.0225 | 3.7980 | 5.4231 | 0.8967 |
| EDSR [14] | 8 | 38.6719 | 0.9359 | 0.0216 | 3.4233 | 5.1530 | 0.9025 |
| 3DFCNN [19] | 8 | 38.8676 | 0.8737 | 0.0210 | 3.4619 | 5.2309 | 0.9047 |
| SSPSR [45] | 8 | 39.1889 | 0.8829 | 0.0204 | **3.2637** | 4.9324 | 0.9127 |
| MCNet [20] | 8 | 39.1702 | 0.8819 | 0.0207 | 3.5117 | 4.9475 | 0.9123 |
| ERCSR [21] | 8 | <u>39.3185</u> | <u>0.8839</u> | **0.0202** | 3.4649 | <u>4.8799</u> | <u>0.9144</u> |
| SFCSR [47] | 8 | 39.2607 | 0.8829 | <u>0.0203</u> | 3.4426 | 4.9007 | 0.9137 |
| F3DUN | 8 | **39.3265** | **0.8849** | 0.0204 | <u>3.2739</u> | **4.8552** | **0.9149** |

**Table 6.** Comparison of computational complexity among the proposed F3DUN and state-of-the-art methods. All metrics are measured for ×4 upscaling on the CAVE dataset.

| Method | EDSR [14] | 3DFCNN [19] | SSPSR [45] | MCNet [20] | ERCSR [21] | SFCSR [47] | F3DUN |
|---|---|---|---|---|---|---|---|
| Params × $10^6$ | 1.54 | 0.04 | 12.89 | 2.17 | 1.59 | 1.22 | 2.40 |
| FLOPs × $10^9$ | 36.98 | 328.99 | 977.50 | 4483.56 | 4460.10 | 2011.9 | 4932.08 |

To compare the reconstruction effect more straightforwardly, we also present some visual results. Figure 8 shows the reconstruction results and absolute error maps with the ×4 upsampling of some bands in the CAVE dataset. The results shown in the figure are a reconstructed single spectral band (represented in grayscale), and all images are normalized. Generally, the images generated by our model have a lower error, and spatial details are recovered better. Specifically, our F3DUN could better recover sharp edges such as those in the red rectangles in Figure 8. As is seen in the third line in Figure 8, F3DUN handles images with higher contrast successfully, while the other methods have the problem that the overall brightness is too dark or too light. Some super-resolution results and error maps with ×4 upsampling in the Harvard dataset are shown in Figure 9. Obviously, our method is better at recovering complex contexts, such as the lattice of the chair (line 3). In Figure 10, we show the reconstructed results and absolute error maps with ×8 upsampling of some bands in the CAVE dataset and Harvard dataset. From the seventh row, we can see that different degrees of deformation and artifacts appear in the reconstruction results of several other deep learning methods. In Figure 11, we visualize the spectrum curves at the pixel positions of (65, 125), (250, 250), and (340, 340) for *chart_and_stuffed_toy* in the CAVE dataset. Among all the methods, the reconstructed spectrum vectors are quite

similar at each location. Though the difference is tiny, the curve of EDSR has an observable deviation from the ground truth, while the spectral distortion of our model is minimal. Similar phenomena also occur in the Harvard dataset, the results of which are shown in Figure 12.

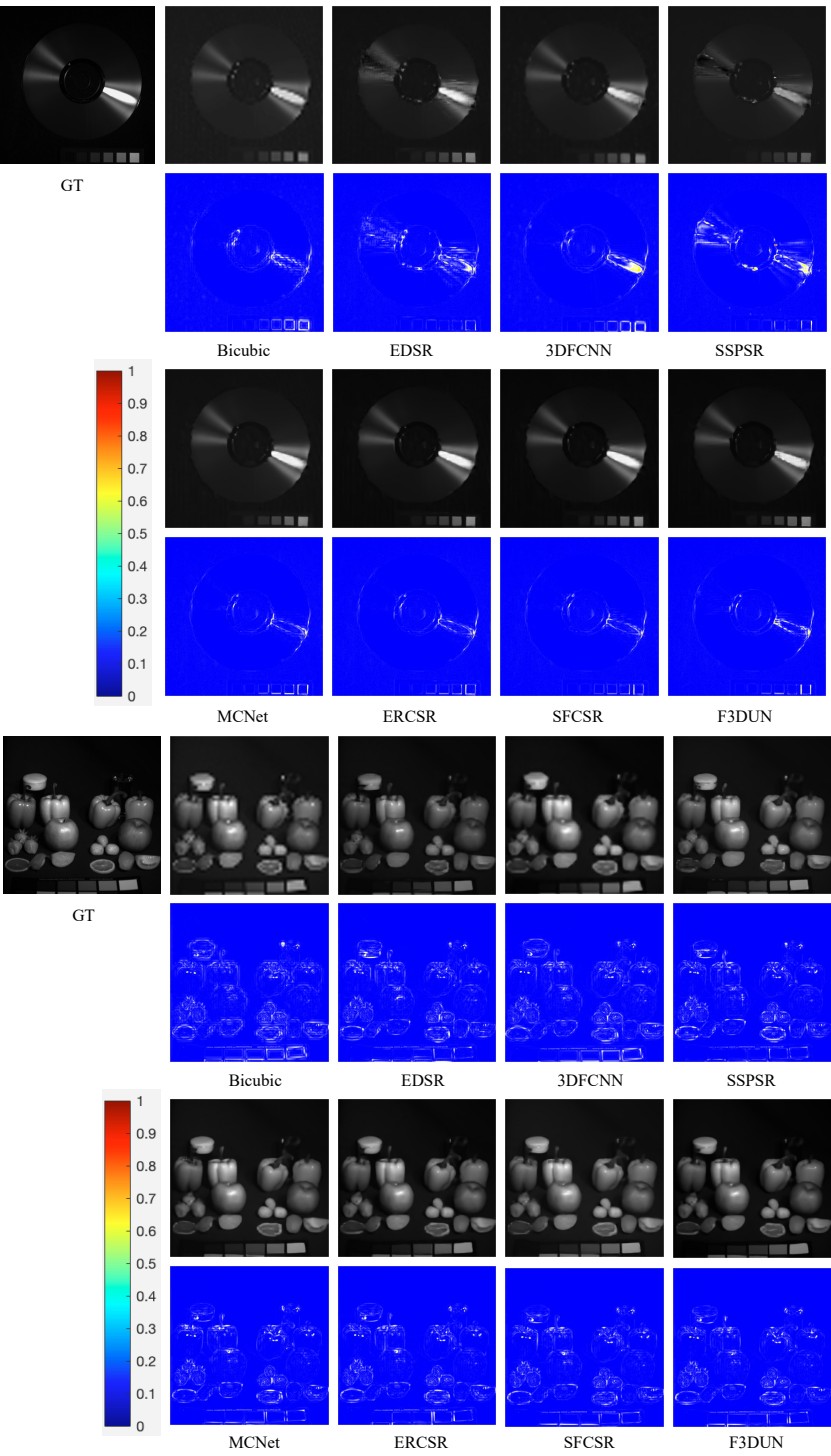

**Figure 8.** Reconstruction results and error maps in the CAVE dataset. From top to bottom: *cd_ms* at 530 nm (4×), *fake_and_real_food_ms* at 690 nm (4×).

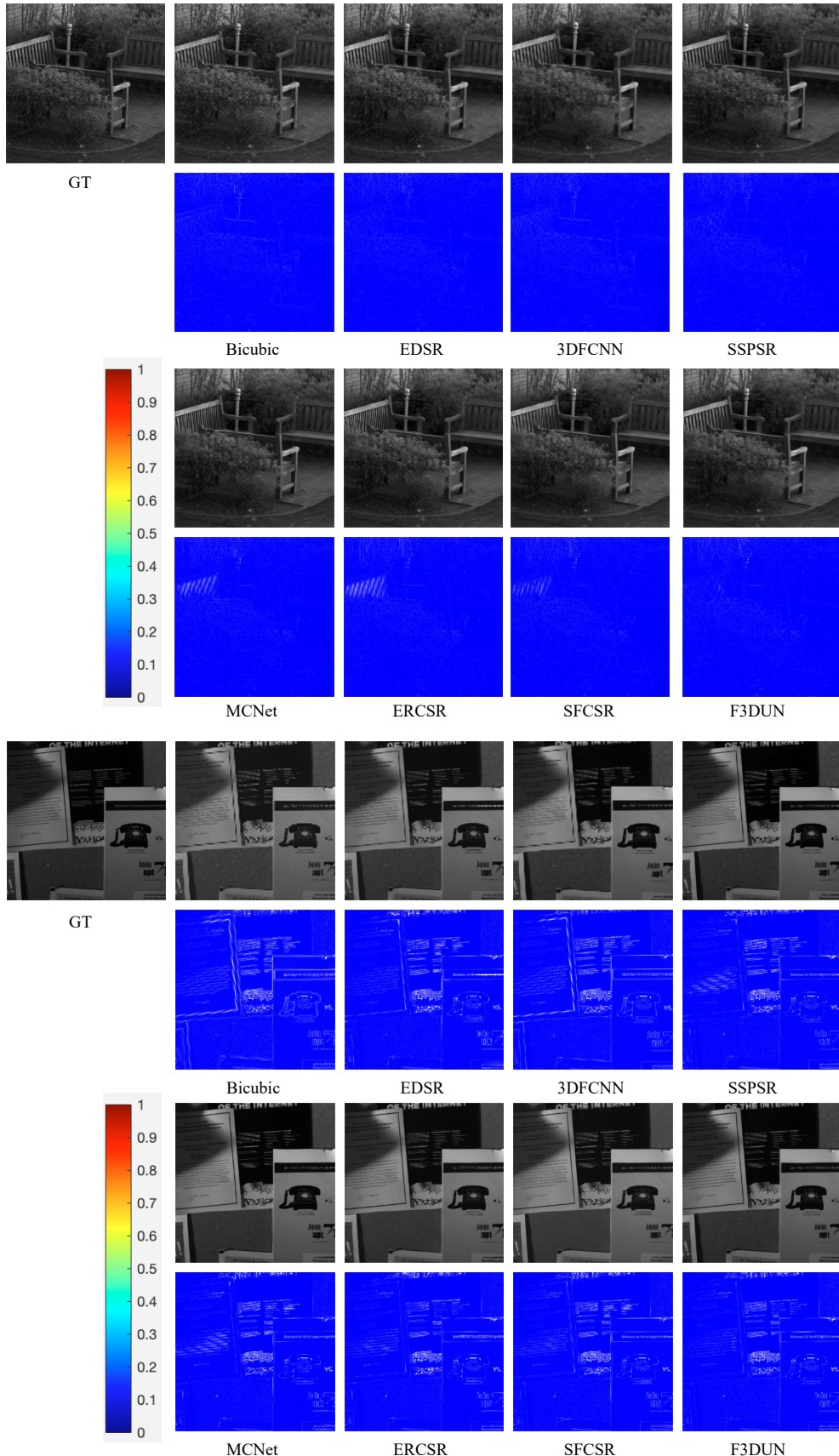

**Figure 9.** Reconstruction results and error maps in the Harvard dataset. From top to bottom: *img4* at 690 nm (4×), *imga6* at 560 nm (4×).

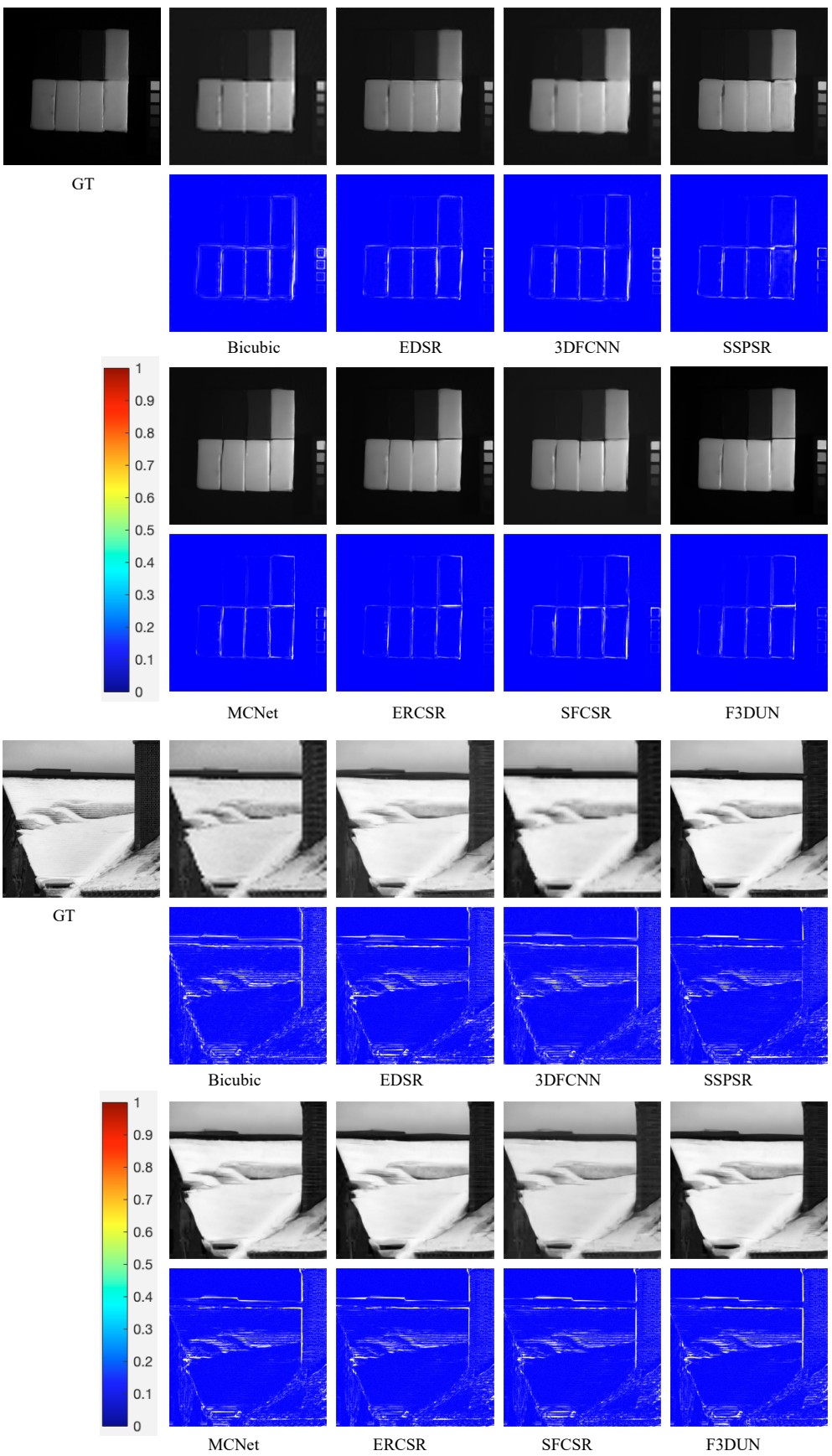

**Figure 10.** Reconstruction results and error maps in CAVE dataset and Harvard dataset. From top to bottom: *clay_ms* at 670 nm (8×) from CAVE dataset and *img4* at 640 nm (8×) from Harvard dataset.

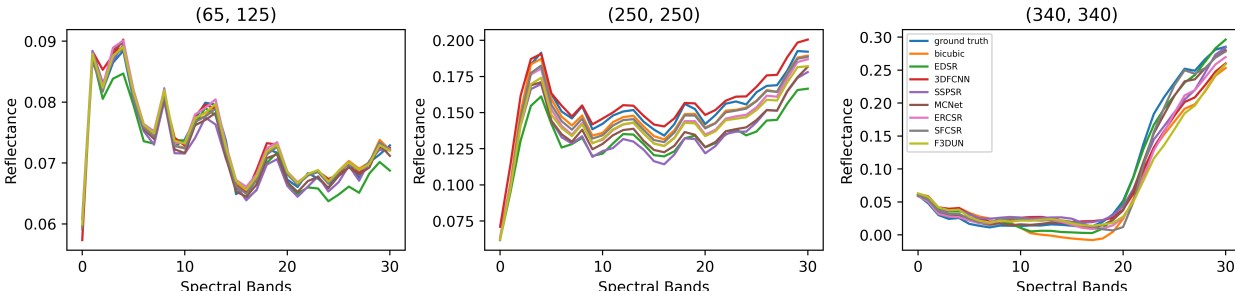

**Figure 11.** Visual comparison of spectral curves for image *chart_and_stuffed_toy* at three pixel positions in the CAVE dataset (4×).

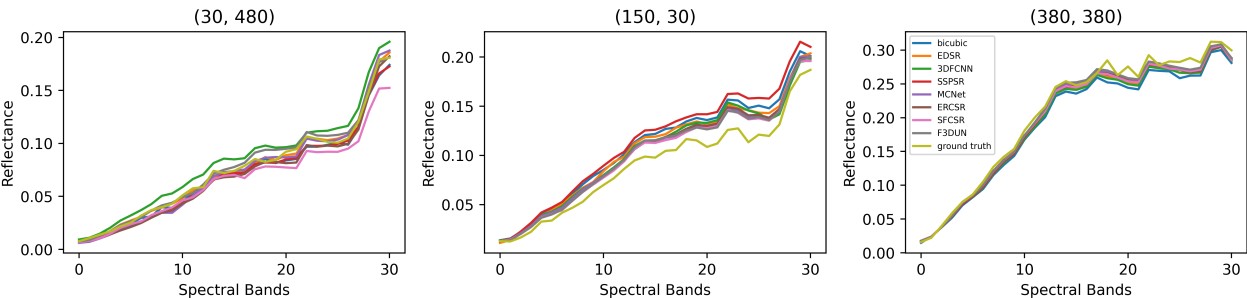

**Figure 12.** Visual comparison of spectral curves for image *img4* at three pixel positions in the Harvard dataset (4×).

## 5. Conclusions

In this paper, we rethink an important model prior for HSISR, namely, the full 3D CNN model, in view of model design. We argue that this model prior is not inevitable, and it could achieve good performance with combination of advanced architectures. Thus, we combine the full 3D CNN model with the U-Net architecture, introducing skip connections to save memory and using a multi-scale structure. Extensive experiments show that the full 3D CNN model can achieve a performance that surpasses the current single-image hyperspectral image super-resolution methods and rediscovers the effectiveness of full 3D models for this task. Through comparison with MUN, the 3D/2D mixed counterpart of F3DUN, we find that 3D convolutions can bring higher performance improvements than 2D ones using the same number of parameters; that is, the modeling capacity of the full 3D model exceeds that of the 3D/2D mixed model. Furthermore, contrary to common belief, the full 3D CNN model can work in a setting with fewer training samples, and the requirements of the full 3D model on training data are not as demanding as one might imagine. All these illustrate that, combined with more advanced network structures, the full 3D model can be a simple and effective method to play a greater role in the field of hyperspectral image super-resolution.

In the future, we plan to explore a more lightweight full 3D CNN for hyperspectral image super-resolution in order to reduce the computational workload. There are some successful works for 2D convolution, such as separable convolution [78] and dilated convolution [79], but lightweight 3D convolution still needs more effort. In addition, some useful designs, such as an attention mechanism, can be combined with full 3D models. Our experiments prove the advancement of full 3D models in hyperspectral image super-resolution, and we believe F3DUN opens a new door for more 3D CNN-based methods to come.

**Author Contributions:** Conceptualization, J.J., Methodology, Z.L.; Writing and original draft preparation, Z.L., W.W. and Q.M.; Writing, review, and editing, X.L. and J.J.; Supervision: J.J. All authors have read and agreed to the published version of the manuscript.

**Funding:** The research was supported by the National Natural Science Foundation of China (61971165, 92270116).

**Data Availability Statement:** The CAVE dataset used in this study is accessible from https://www.cs.columbia.edu/CAVE/databases/multispectral, and the Harvard dataset used in this study is accessible from http://vision.seas.harvard.edu/hyperspec/index.html.

**Conflicts of Interest:** The authors declare no conflict of interest.

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
