# Peer review of "Rethinking 3D-CNN in Hyperspectral Image Super-Resolution"

_remotesensing, doi:10.3390/rs15102574_

Round 1

Reviewer 1 Report

The authors have proposed a develop a 3D/2D mixed model named F3DUN for hyperspectral video tracking. The manuscript is complete, and the authors try to prove the progressiveness of the algorithm through experiments. However, there are some problems that need to be revised. The comments are as follows

1.      First, the computational complexity of the algorithm needs to be analyzed and compared with SOTA algorithm.

2.      I suggest reducing the contribution of the paper to three.

3.      Need to add ablation experiments in the manuscript.

4.      The references used in the paper are relatively old, so it is recommended to update them. In addition, some more methods regarding remote sensing using graph-based methods should be investigated in your introduction, e.g., Semi-Supervised Locality Preserving Dense Graph Neural Network With ARMA Filters and Context-Aware Learning, Unsupervised Self-correlated Learning Smoothy Enhanced Locality Preserving Graph Convolution Embedding Clustering, Self-supervised Locality Preserving Low-pass Graph Convolutional Embedding, AF2GNN: Graph Convolution with Adaptive Filters and Aggregators, Multi-feature Fusion: Graph Neural Network and CNN Combining, MultiReceptive Field: An Adaptive Path Aggregation Graph Neural Framework.

5.      How about the adaptability of the algorithm to different number of training labels, especially small labels. Please compare with the SOAT methods.

No more comments.

Author Response

Responses to Reviewer #1

       The authors have proposed a develop a 3D/2D mixed model named F3DUN for hyperspectral video tracking. The manuscript is complete, and the authors try to prove the progressiveness of the algorithm through experiments. However, there are some problems that need to be revised. The comments are as follows.

       Response: We thank the reviewer very much for the supportive comments. We have carefully revised the manuscript based on the comments of the reviewer. Please see the following for more details.

  1. First, the computational complexity of the algorithm needs to be analyzed and compared with SOTA algorithm.

Response: Thanks for the valuable comments. In the revised manuscript, according to your comment, we have conducted some experiments and summarized the computational complexity of different methods in Table 5, and made analysis in Section 4.5 in the re-submitted manuscript. In short, our model lies in the same level of parameters and FLOPs with MCN, but has better performance than SOTA algorithms.

In Table 5, we compare the model scale and computational complexity of our F3DUN and other state-of-the-art methods. It is shown that our model achieves superior performance with little extra computation cost. In particular, the parameter amount and FLOPs of F3DUN are at the same level with these of MCNet, while F3DUN outperforms MCNet on all six quantitive metrics. The main cost of our model comes from its pure 3D CNN structure, since 3D convolutions are very dense on parameter and calculation. Our model is slightly heavier than mixed 3D-2D models (i.e., MCNet and ERCSR), and 3DFCNN is lightweight due to its shallow structure. Though sequence models like SFCSR have less parameters, they are very slow in reference stage.

  1. I suggest reducing the contribution of the paper to three.

Response: Thank you for your valuable suggestion. In the revised manuscript, we have rewritten the contribution of the paper.

We rethink the role that 3D CNN plays in the HSISR field, and design a novel full 3D CNN model based on U-Net architecture called Full 3D U-Net (F3DUN). Experimentally it outperforms existing state-of-the-art single-image SR methods, which proves the effectiveness of full 3D CNN model in this field.

We develop a mixed 3D/2D model that shares the same structure with F3DUN, termed as Mixed U-Net (MUN) for comparison. Extensive analysis on the two models show that the full 3D CNN model has larger model capacity than the 3D/2D mixed model with the same parameter number, thus it performs better with large-scale datasets.

We explore the relationship between the scale of training samples and the prior of the model. And we argue that the full 3D CNN model can get competitive results on small-scale training sets compared with the 3D/2D mixed model, which concludes that the full 3D CNN model is more robust with respect to the amount of training samples than common sense.

  1. Need to add ablation experiments in the manuscript.

Response: Thank you for your valuable comment. In the revised manuscript, according to your comments, we have added ablation experiments in the manuscript. The proposed F3DUN has simple U-Net architecture and we don’t introduce special modules other than 3D convolution. Skip connections has been widely used in deep learning, and in Section 4.4, we have proven that full 3D model is a better prior than 3D/2D mixed model.

Hyperspectral images are easy to suffer various distortions. However, the robustness to degradation is not our main concern. On the one hand, we argue that benchmarks are collected by real-world sensors, so the performance on them is meaningful to practical cases. On the other hand, super-resolution is a subset of hyperspectral image reconstructions, and there are many techniques that can deal with various degradations, like denoising [74] [75] and deblurring [76] [77].

In Fig. A1, and Fig. A2, we present reconstruction results with noisy inputs on the CAVE dataset. It can be seen that our model is somewhat robust to gaussian noise, but may be severely affected by impulse noise. The main reason is that we don't introduce impulse noisy samples in training.

  1. The references used in the paper are relatively old, so it is recommended to update them. In addition, some more methods regarding remote sensing using graph-based methods should be investigated in your introduction, e.g., Semi-Supervised Locality Preserving Dense Graph Neural Network With ARMA Filters and Context-Aware Learning, Unsupervised Self-correlated Learning Smoothy Enhanced Locality Preserving Graph Convolution Embedding Clustering, Self-supervised Locality Preserving Low-pass Graph Convolutional Embedding, AF2GNN: Graph Convolution with Adaptive Filters and Aggregators, Multi-feature Fusion: Graph Neural Network and CNN Combining, MultiReceptive Field: An Adaptive Path Aggregation Graph Neural Framework.

Response: Thanks for the valuable suggestion. In the revised manuscript, we have refreshed references.

The former introduces multiple 2D branches to extract multiscale spatial information from the feature generated by 3D convolutions in each block (see Fig. 1d), and the latter adds one 2D unit after every 3D unit to concentrate on spatial features (Fig. 1c). Recently, graph-based methods are applied on many tasks in hyperspectral image processing, such as classification [22] [23] [24], clustering [25] [26] [27], target detection [28] and anomaly detection [29], and achieve good performance.

  1. How about the adaptability of the algorithm to different number of training labels, especially small labels. Please compare with the SOAT methods.

Response: Thank you for your valuable comment. We have revised our manuscript to address your concern. In Section 4.4, we show that the proposed F3DUN outperforms MUN on the reduced CAVE dataset.

The performances of MUN and F3DUN on the reduce dataset are shown in Table 2. With fewer training samples, both models have a certain drop on most metrics, which is consistent with the common sense of deep learning-based models. However, the full 3D CNN model still achieves acceptable results on this smaller dataset, even outperforming the 3D/2D mixed model on several metrics such as PSNR, RMSE, and ERGAS. This demonstrates that the full 3D CNN model is less sensitive to the scale of training samples, against what people commonly thought, and it could achieve competitive performance compared with the 3D/2D mixed model on the small-scale dataset.

Reviewer 2 Report

The authors have proposed a a 3D/2D mixed method. The manuscript is complete, and the authors try to prove the progressiveness of the algorithm through experiments. However, there are some problems that need to be revised. The comments are as follows

1. The motivations or remaining challenges are not so clear or what kinds of issues or difficulties are this task that is facing. Please give more details and discussion about the key problems solved in this paper, which is largely different from existing works.

2. A deep literature review should be given, particularly advanced and SOTA deep learning or AI models in hyperspectral image classification. Therefore, the reviewer suggests discussing some currently SOTA works by analyzing the following papers in the revised manuscript, e.g., Multi-scale Receptive Fields: Graph Attention Neural Network, Global to Local: A Hierarchical Detection Algorithm for Hyperspectral Image Target Detection.

3. How about the computational complexity?

4. The compared methods are not sufficient. Some SOTA compared methods should be involved.

5. It is well-known that the hyperspectral image usually tend to suffer from various degradation, noise effects, or variabilities in the process of imaging. Please give the discussion and analysis by referring to the paper titled by e.g., A Novel Hyperspectral Image Classification Model Using Bole Convolution With Three-Direction Attention Mechanism: Small Sample and Unbalanced Learning. The reviewer is wondering what will happen if the proposed method meets the various variabilities.

6. Some more future directions should be pointed out in the conclusion.

7.What is the adaptability of the algorithm proposed by the authors to image noise? Please use experiments to prove the progressiveness of the algorithm. That is to say, what is the classification performance of the algorithm when images are injected with different noises. In addition, when the classes are unbalanced, what is the classification effect of the algorithm.

No more comments.

Author Response

Responses to Reviewer #2

       The authors have proposed a 3D/2D mixed method. The manuscript is complete, and the authors try to prove the progressiveness of the algorithm through experiments. However, there are some problems that need to be revised. The comments are as follows.

       Response: We thank the reviewer very much for the supportive comments. We have carefully revised the manuscript based on the comments of the reviewer. Please see the following for more details.

  1. The motivations or remaining challenges are not so clear or what kinds of issues or difficulties are this task that is facing. Please give more details and discussion about the key problems solved in this paper, which is largely different from existing works.

Response: Thanks for the valuable suggestions. In the revised manuscript, according to your comment, we have described our motivation in Section 1.

However, existing 3D HSISR models are very simple and plain, and they didn’t combine with some advanced inventions of deep learning. Therefore, it is worth questioning whether the critical views on 3D models still hold. And we believe that the potential of 3D models has not been fully explored yet. These drive us to revisit those structures that have been proven effective in the super-resolution field and combine them with 3D CNN.

Moreover, in Section 3.4, we explain the difference among our F3DUN and two typical model priors of HSISR, F3DCNN and 3D/2D mixed models.

(a) F3DCNN: F3DCNN is the pioneer work of full 3D CNN models for HSISR. It consists of 5 3D convolution layers, and uses MSE loss to train. However, there are noticeable drawbacks to it. Most importantly, the model architecture of F3DCNN is too shallow and it does not combine with advanced ideas of deep learning, such as residual learning and long skip connection. Therefore, in this paper, we rethink the prior of full 3D CNN in HSISR, and argue that full 3D CNN model’s bad results are not caused by 3D CNN, but unreasonable model design. It is obvious that F3DUN is much deeper than 3DFCNN, but successfully prevents overfitting and achieves SOTA results.

(b) 3D/2D mixed models: The main idea of 3D/2D mixed models is to introduce 2D convolutions into 3D CNN to boost the ability of spatial enhancement. They have become popular recently and can be seen as a modification for full 3D CNN models. However, there are two potential problems. On the one hand, the 2D convolutions in 3D/2D models are shared on the spectral dimension, which leads to a risk of spectral distortion. On the other hand, 3D/2D mixed models always face a dilemma to balance these two kinds of convolutions, where features enhanced by 2D convolutions could be polluted by cascading 3D modules. In MUN, we partially solve the two above problems, but we make a step forward: could a full 3D CNN model outperform 3D/2D mixed ones? Model analysis on MUN and F3DUN supports our idea and proves the advantages of the full 3D CNN model compared with 3D/2D mixed models.

  1. A deep literature review should be given, particularly advanced and SOTA deep learning or AI models in hyperspectral image classification. Therefore, the reviewer suggests discussing some currently SOTA works by analyzing the following papers in the revised manuscript, e.g., Multi-scale Receptive Fields: Graph Attention Neural Network, Global to Local: A Hierarchical Detection Algorithm for Hyperspectral Image Target Detection.

Response: Thank you for your valuable suggestion. We have added some discussion on recent development of hyperspectral image processing in Section 1, and supplement advanced models in Section 2.

The latter comment leads to 3D/2D mixed models, which introduce extra 2D convolutions to boost spatial details. Representatives of such methods are MCNet [16] and ERCSR [17]. The former introduces multiple 2D branches to extract multiscale spatial information from the feature generated by 3D convolutions in each block (see Fig. 1d), and the latter adds one 2D unit after every 3D unit to concentrate on spatial features (Fig. 1c). Recently, graph-based methods are applied on many tasks in hyperspectral image processing, such as classification [18] [19] [20], clustering [21] [22] [23], target detection [24] and anomaly detection [25], and achieve good performance.

Recently, Liu et al. [44] proposed a parallel network called Interactformer for SHISR, which contains a Transformer branch and a 3D-CNN branch. The Transformer branch is used to capture long-range dependencies to obtain global features, and the 3-D CNN branch is used to extract local features while preserving the spectral correlation of HSIs.

  1. How about the computational complexity?

Response: Thanks for the valuable comments. In the revised manuscript, according to your comment, we have summarized related information in Table 5, and made analysis in Section 4.5 in the re-submitted manuscript. In short, our model lies in the same level of parameters and FLOPs with MCN, but has better performance than SOTA algorithms.

In Table 5, we compare model scale and computational complexity of our F3DUN and other state-of-the-art methods. It is shown that our model achieves superior performance with little extra computation cost. In specific, the parameter amount and FLOPs of F3DUN are at the same level of MCNet, while F3DUN outperforms MCNet on all six quantitive metrics. And the main cost of our model comes from its pure 3D CNN structure, since 3D convolutions are very dense on parameter and calculation. Our model is slightly heavier than mixed 3D-2D models (i.e. MCNet and ERCSR), and 3DFCNN is lightweight due to its shallow structure. Though sequence models like SFCSR have less parameters, they can be slow in running due to bad parallelism.

  1. The compared methods are not sufficient. Some SOTA compared methods should be involved.

Response: Thank you for your valuable comment. In Section 4.5, we have compared the most related HSISR models based on CNN. The main novelty of this paper is to rethink the role of 3D CNN model in HSISR field, and the comparison covers majority of CNN-based SOTA methods. We do notice that there are some newly-published works, but most of them using different architecture like transformer. Consequently, we think that our comparison is sufficient to support our idea.

  1. It is well-known that the hyperspectral image usually tend to suffer from various degradation, noise effects, or variabilities in the process of imaging. Please give the discussion and analysis by referring to the paper titled by e.g., A Novel Hyperspectral Image Classification Model Using Bole Convolution With Three-Direction Attention Mechanism: Small Sample and Unbalanced Learning. The reviewer is wondering what will happen if the proposed method meets the various variabilities.

Response: It is true as the reviewer suggested that hyperspectral images often suffer various degradation. In Section 1, we add some analysis referring the paper that the reviewer suggested. And it’s worthy noticing that super-resolution is a subset of hyperspectral image reconstruction, and it only tackle one degradation, i.e. downsampling. And there are many other techniques to deal with other kinds of degradation, such as denoising and deblurring.

The problem caused by high dimensionality of HSI is severe. In [ 15], Cai et. al tried to solve this issue with 3D attention mechanism and information bottleneck for classification task. However, in super-resolution, the model needs to preserve sufficient information of the input, thus a natural way to solve this problem is to introduce 3D convolutions, which is good at extracting spatial-spectral correlations.

  1. Some more future directions should be pointed out in the conclusion.

Response: Thanks for the comment. Following suggestions of the viewer, we have updated the conclusion part in the revised manuscript. In short, there are two main future directions: one is lightweight 3D model, the other is introducing more advanced modules like attention.

In the future, we plan to explore more lightweight full 3D CNN for hyperspectral image super-resolution, in order to reduce computation workload. There are some successful works for 2D convolution, such as separable convolution [ 73 ] and dilated convolution [74], but lightweight 3D convolution still needs more effort. And some useful designs like attention mechanism can be combined with full 3D models. Our experiments prove the advance of full 3D model in hyperspectral image super-resolution, and we believe F3DUN opens a new gate to more 3D CNN-based methods to come.

  1. What is the adaptability of the algorithm proposed by the authors to image noise? Please use experiments to prove the progressiveness of the algorithm. That is to say, what is the classification performance of the algorithm when images are injected with different noises. In addition, when the classes are unbalanced, what is the classification effect of the algorithm.

Response: Thanks to the comment. We have added some experimental results with noisy inputs in Appendix B. It can be seen that our model is more robust to gaussian noise than impulse noise. However, it is natural that our model isn’t so robust to noise, because we do not add any noise in training. Basically, denoising is beyond our scope, since noise can be removed by denoising techniques in preprocessing. On the other hand, since the two benchmark are collected by real-world sensors, our result is meaningful in practical cases. As for unbalanced label problem, our model aims to super-resolution, so it is not our concerns.

Hyperspectral images are easy to suffer various distortions. However, the robustness to degradation is not our main concern. On the one hand, we argue that benchmarks are collected by real-world sensors, so the performance on them is meaningful to practical cases. On the other hand, super-resolution is a subset of hyperspectral image reconstructions, and there are many techniques that deals with degradation, like denoising [74] [75] and deblurring [76] [77].

In Fig. A1, and Fig. A2, we present reconstruction results with noisy inputs on the CAVE dataset. It can be seen that our model is somewhat robust to gaussian noise, but can be severely affected by impulse noise. The main reason is that we don't introduce noise manually in training, and the model has no chance to learn to remove it.

Reviewer 3 Report

In this paper the authors present full 3D CNN model and combine it with U-net architecture introducing skip connection to save memory. They found that 3Dconvolutions can bring higher performance improvement over the 2D ones at same parameter level and the model capacity of the 3D model has advantages over the 2D model.   First of all, the weak point of this article is precisely the way it is written, which jeopardizes its following by readers, especially in a text of this complexity. A careful revision of the text is suggested, in order to use shorter sentences (some of the current ones should be divided), use commas and periods judiciously to clearly separate the ideas presented, and pay some attention to grammatical details. For example on line 82 fig should be Fig.Also on line 320-322 there must be space between the method ad the citation. I have some general recommendations and comments. I recommend to add a chart of methodology in Section 3. I also advice to add a separate Table with all abbreviations used in the text for easier reading. General recommendations for the Figures is that they are very small, you should increase them for easier reading, this is necessary especially for Figure 3 and 4. The images on Figure 5 are almost impossible to be seen, suggest increasing Figure 5. The same is true for Figure 6. When you talk about mathematical modeling of neural network I recommend to cite some more recent papers in the field: Doicu, A.; Doicu, A.; Efremenko, D.S.; Loyola, D.; Trautmann, T. An Overview of Neural Network Methods for Predicting Uncertainty in Atmospheric Remote Sensing. Remote Sens. 2021, 13, 5061. https://doi.org/10.3390/rs13245061 Todorov, V. (2022). Advanced Monte Carlo Methods to Neural Networks. In: Fidanova, S. (eds) Recent Advances in Computational Optimization. WCO 2021. Studies in Computational Intelligence, vol 1044. Springer, Cham. https://doi.org/10.1007/978-3-031-06839-3_17 Zhu, L.; Huang, L.; Fan, L.; Huang, J.; Huang, F.; Chen, J.; Zhang, Z.; Wang, Y. Landslide Susceptibility Prediction Modeling Based on Remote Sensing and a Novel Deep Learning Algorithm of a Cascade-Parallel Recurrent Neural Network. Sensors 2020, 20, 1576. https://doi.org/10.3390/s20061576 Dimov, I.,et al., V. An unbiased Monte Carlo method to solve linear Volterra equations of the second kind. Neural Comput & Applic 34, 1527–1540 (2022). https://doi.org/10.1007/s00521-021-06417-5 On Tables 3 and 4 you should point why in some cases SSPSR approach is better. How this method compared to your method in terms of computational complexity and convergence? The benefits of the proposed technique should be highlighted in the Discussion compared with other well known approaches. If possible discuss the computational complexity of the proposed methodology and the other methods included in the study EDSR, 3DFCNN, SSPSR, MCNet, ERCSR, SFCSR, F3DUN. How the computational complexity of your method is compared to other state of art algorithms. The paper does not provide any discussion of the limitations of the proposed method or any suggestions for improvement. The authors also do not provide any discussion of the implications of the results or any suggestions for further research. These points should be addressed before the paper is published. It is also not clear what the software that was used for the algorithm implementation and computational issues. The Conclusion is too short, I advice to separate some highlights of your work as most important and novel achievements. I recommend address this issues and accepting after major revision.

Extensive editing of the text is required. I only mention some errors for example.
Watch out for proper sentences, missing space and commas, capital letters, pass tenses.
A careful revision of the text is suggested, in order to use shorter sentences
(some of the current ones should be divided),
use commas and periods judiciously to clearly separate the ideas presented,
and pay some attention to grammatical details.
For example on line 82 fig should be Fig.Also on line 320-322 there must be space between the method ad the citation.

Author Response

Responses to Reviewer #3

       In this paper the authors present full 3D CNN model and combine it with U-net architecture introducing skip connection to save memory. They found that 3Dconvolutions can bring higher performance improvement over the 2D ones at same parameter level and the model capacity of the 3D model has advantages over the 2D model.

       Response: We are very grateful to the reviewer for the supportive comments. We have carefully adjusted the manuscript based on the comments of the reviewer. Please see the following for more details.

  1. First of all, the weak point of this article is precisely the way it is written, which jeopardizes its following by readers, especially in a text of this complexity. A careful revision of the text is suggested, in order to use shorter sentences (some of the current ones should be divided), use commas and periods judiciously to clearly separate the ideas presented, and pay some attention to grammatical details. For example on line 82 fig should be Fig.Also on line 320-322 there must be space between the method ad the citation. I have some general recommendations and comments. I recommend to add a chart of methodology in Section 3. I also advice to add a separate Table with all abbreviations used in the text for easier reading. General recommendations for the Figures is that they are very small, you should increase them for easier reading, this is necessary especially for Figure 3 and 4. The images on Figure 5 are almost impossible to be seen, suggest increasing Figure 5. The same is true for Figure 6.

Response: Thank you for the detailed comment. Considering the reviewer’s suggestion, we have adjusted the structure and sentences in introduction and conclusion. And we carefully check grammar and format errors.

However, existing 3D HSISR models are very simple and plain, and they didn’t combine with some advanced inventions of deep learning. Therefore, it is worth questioning whether the critical views on 3D models still hold. And we believe that the potential of 3D models has not been fully explored yet. That drives us to revisit those structures that have been proven effective in the super-resolution field and combine them with 3D CNN. Specifically, we propose a simple yet effective full 3D CNN model with U-Net architecture, called Full 3D U-Net (F3DUN). F3DUN replaces 2D blocks in normal U-Net with 3D convolutions and removes the downsampling-upsampling process in the middle. Experiments show that F3DUN outperforms state-of-the-art methods, which proves effectiveness of full 3D model in HSISR task.

Comparison on a small-scale dataset overthrows this point, since F3DUN still gets competitive results with MUN. With little harm on performance, F3DUN eliminates severe overfitting.

In Section 3, Fig. 3 and Fig. 4 are the charts for F3DUN and MUN respectively. And we add a table for abbreviations in Appendix A. As for figures, a band only contains limited information of a narrow range of wavelight, that’s why most of results seem dark.

  1. When you talk about mathematical modeling of neural network I recommend to cite some more recent papers in the field: Doicu, A.; Doicu, A.; Efremenko, D.S.; Loyola, D.; Trautmann, T. An Overview of Neural Network Methods for Predicting Uncertainty in Atmospheric Remote Sensing. Remote Sens. 2021, 13, 5061. https://doi.org/10.3390/rs13245061 Todorov, V. (2022). Advanced Monte Carlo Methods to Neural Networks. In: Fidanova, S. (eds) Recent Advances in Computational Optimization. WCO 2021. Studies in Computational Intelligence, vol 1044. Springer, Cham. https://doi.org/10.1007/978-3-031-06839-3_17 Zhu, L.; Huang, L.; Fan, L.; Huang, J.; Huang, F.; Chen, J.; Zhang, Z.; Wang, Y. Landslide Susceptibility Prediction Modeling Based on Remote Sensing and a Novel Deep Learning Algorithm of a Cascade-Parallel Recurrent Neural Network. Sensors 2020, 20, 1576. https://doi.org/10.3390/s20061576 Dimov, I.,et al., V. An unbiased Monte Carlo method to solve linear Volterra equations of the second kind. Neural Comput & Applic 34, 1527–1540 (2022). https://doi.org/10.1007/s00521-021-06417-5.

Response: The reviewer’s comment shows good knowledge and insights. And we cited some papers as the reviewer suggests in Section 1, to provide a big picture of deep-learning in remote sensing. But it’s worthy mentioning that our model is simple and purely conducted with 3D convolutions, which relates little of mathematical modeling of neural network. The main novelty of this work is to propose a simple and effective model prior of HSISR, and shows good potential of full 3D CNN on this task.

However, the mainstream of natural image SR methods is 2D CNN, which could lead to severe spectral distortion when applied to multi-band HSIs. Recently, there emerges some methods that combines hand-craft prior and neural networks, and introduce mathematical modeling considering properties of HSI, such as [15], [16], [17].

  1. On Tables 3 and 4 you should point why in some cases SSPSR approach is better.

Response: Thank you for your valuable comment. We have added relative analysis in Section 4. We believe the divide-and-merge architecture helps SSPSR preserve spectral information better.

It can be seen that SSPSR achieves sub-optimal scores for most indicators on the Harvard dataset with respect to 4X SR. SSPSR shows slight advantage on SAM, and this margin possibly comes from its divide-and-merge strategy. Since SSPSR first process each band separately and then merge all bands together, this architecture could preserve spectral information.

  1. How this method compared to your method in terms of computational complexity and convergence? The benefits of the proposed technique should be highlighted in the Discussion compared with other well known approaches. If possible discuss the computational complexity of the proposed methodology and the other methods included in the study EDSR, 3DFCNN, SSPSR, MCNet, ERCSR, SFCSR, F3DUN. How the computational complexity of your method is compared to other state of art algorithms.

Response: Thanks for the valuable comments. In the revised manuscript, according to your comment, we have summarized related information in Table 5, and made analysis in Section 4.5 in the re-submitted manuscript. In short, our model lies in the same level of parameters and FLOPs with MCN, but has better performance than SOTA algorithms.

In Table 5, we compare model scale and computational complexity of our F3DUN and other state-of-the-art methods. It is shown that our model achieves superior performance with little extra computation cost. In specific, the parameter amount and FLOPs of F3DUN are at the same level of MCNet, while F3DUN outperforms MCNet on all six quantitive metrics. And the main cost of our model comes from its pure 3D CNN structure, since 3D convolutions are very dense on parameter and calculation. Our model is slightly heavier than mixed 3D-2D models (i.e. MCNet and ERCSR), and 3DFCNN is lightweight due to its shallow structure. Though sequence models like SFCSR have less parameters, they can be slow in running due to bad parallelism.

  1. The paper does not provide any discussion of the limitations of the proposed method or any suggestions for improvement. The authors also do not provide any discussion of the implications of the results or any suggestions for further research. These points should be addressed before the paper is published. It is also not clear what the software that was used for the algorithm implementation and computational issues. The Conclusion is too short, I advice to separate some highlights of your work as most important and novel achievements.

Response: Thanks for the comment. Following suggestions of the viewer, we have updated the conclusion part in the revised manuscript. In short, there are two main future directions: one is lightweight 3D model, the other is introducing more advanced modules like attention.

In the future, we plan to explore more lightweight full 3D CNN for hyperspectral image super-resolution, in order to reduce computation workload. There are some successful works for 2D convolution, such as separable convolution [ 73 ] and dilated convolution [74], but lightweight 3D convolution still needs more effort. And some useful designs like attention mechanism can be combined with full 3D models. Our experiments prove the advance of full 3D model in hyperspectral image super-resolution, and we believe F3DUN opens a new gate to more 3D CNN-based methods to come.

Round 2

Reviewer 1 Report

No more comments.

Reviewer 2 Report

The paper can be accepted in present form.

Reviewer 3 Report

Thanks the authors that they addressed all my comments and questions so now I recommend the paper for acceptance.

All my comments for English language are addressed so

I reccommend only minor check at final stage..